# An Exploratory Study on Seasonal Variation in the Gut Microbiota of Athletes: Insights from Japanese Handball Players

**DOI:** 10.3390/microorganisms12040781

**Published:** 2024-04-11

**Authors:** Kazuya Toda, Shin Yoshimoto, Keisuke Yoshida, Eri Mitsuyama, Noriyuki Iwabuchi, Koji Hosomi, Takayuki Jujo Sanada, Miyuki Tanaka, Hinako Nanri, Jun Kunisawa, Toshitaka Odamaki, Motohiko Miyachi

**Affiliations:** 1Innovative Research Institute, Morinaga Milk Industry Co., Ltd., Zama 252-8583, Kanagawa, Japan; kazuya-toda983@morinagamilk.co.jp (K.T.); shin-yoshimoto923@morinagamilk.co.jp (S.Y.); keisuke-yoshida826@morinagamilk.co.jp (K.Y.); e-inoue@morinagamilk.co.jp (E.M.); n-iwabuchi@morinagamilk.co.jp (N.I.); m_tanaka@morinagamilk.co.jp (M.T.); 2Laboratory of Vaccine Materials and Laboratory of Gut Environmental System, Microbial Research Center for Health and Medicine, National Institutes of Biomedical Innovation, Health and Nutrition, Ibaraki 567-0085, Osaka, Japan; hosomi@nibiohn.go.jp (K.H.); jujo-sanada@nibiohn.go.jp (T.J.S.); kunisawa@nibiohn.go.jp (J.K.); 3Department of Respirology, Graduate School of Medicine, Chiba University, Chiba 260-8670, Chiba, Japan; 4Department of Physical Activity Research, National Institutes of Biomedical Innovation, Health and Nutrition, Settsu 566-0002, Osaka, Japan; hnanri@nibiohn.go.jp; 5Laboratory of Gut Microbiome for Health, Microbial Research Center for Health and Medicine, National Institutes of Biomedical Innovation, Health and Nutrition, Ibaraki 567-0085, Osaka, Japan; 6Graduate School of Medicine, Osaka University, Suita 565-0871, Osaka, Japan; 7Graduate School of Pharmaceutical Sciences, Osaka University, Suita 565-0871, Osaka, Japan; 8Graduate School of Dentistry, Osaka University, Suita 565-0871, Osaka, Japan; 9Graduate School of Science, Osaka University, Toyonaka 560-0043, Osaka, Japan; 10International Vaccine Design Center, The Institute of Medical Science, The University of Tokyo, Minato-ku 108-8639, Tokyo, Japan; 11Department of Microbiology and Immunology, Kobe University Graduate School of Medicine, Kobe 650-0017, Hyogo, Japan; 12Faculty of Science and Engineering, Waseda University, Shinjuku-ku 169-0072, Tokyo, Japan; 13Faculty of Sport Sciences, Waseda University, Tokorozawa 359-1192, Saitama, Japan

**Keywords:** gut microbiota, athletes, longitudinal study, alpha-diversity, Japanese male handball players

## Abstract

Despite accumulating evidence that suggests a unique gut microbiota composition in athletes, a comprehensive understanding of this phenomenon is lacking. Furthermore, seasonal variation in the gut microbiota of athletes, particularly during the off-season, remains underexplored. This study aimed to compare the gut microbiotas between athletic subjects (AS) and non-athletic subjects (NS), and to investigate variations between athletic and off-season periods. The data were derived from an observational study involving Japanese male handball players. The results revealed a distinct gut microbiota composition in AS compared with NS, characterized by significantly higher alpha-diversity and a greater relative abundance of *Faecalibacterium* and *Streptococcus*. Moreover, a comparative analysis between athletic and off-season periods in AS demonstrated a significant change in alpha-diversity. Notably, AS exhibited significantly higher alpha-diversity than NS during the athletic season, but no significant difference was observed during the off-season. This study demonstrates the characteristics of the gut microbiota of Japanese handball players and highlights the potential for changes in alpha-diversity during the off-season. These findings contribute to our understanding of the dynamic nature of the gut microbiota of athletes throughout the season.

## 1. Introduction

It is well established that diet [1] and exercise [2] influence the gut microbiota. Numerous studies have suggested that athletes, as a result of their specific diet (e.g., high-protein) and regular physical training, often exhibit a different gut microbiota compared with sedentary subjects [3,4]. Specifically, the gut microbiota of athletes has been reported to have higher alpha-diversity [3] and an abundance of short-chain fatty acid (SCFA)-producing bacteria (e.g., *Faecalibacterium*) [5,6]. Interestingly, several studies have proposed a potential correlation between gut microbiota and exercise performance [7,8]. For example, certain characteristics of the gut microbiota, such as higher alpha-diversity and butyrate-producing bacteria, have been linked with maximal oxygen uptake (VO_2_max), an indicator of aerobic capacity [8]. Additionally, a significant positive correlation between *Bacteroides* and leg extension power has been reported in older Japanese individuals [9]. Therefore, understanding the gut microbiota is crucial because it could potentially influence athletic performance.

Despite these findings, most studies have focused on athletes from Europe, the United States, and China, with limited research on Japanese athletes [1,3,5,6]. Notably, Japanese individuals have been shown to have a unique gut microbiota compared with individuals from China and Western countries [10]. It remains unclear whether the gut microbiota of athletes also exhibits ethnic differences. Furthermore, recent studies have suggested that the gut microbiota may vary among athletes depending on the type of sport (e.g., aerobics, wrestling, rowing) [11]. To address these gaps in knowledge, it is necessary to accumulate microbiome data from athletes of diverse ethnicities from various sports.

Exercise has been suggested to increase the alpha-diversity and the relative abundance of butyrate-producing bacteria in the gut microbiota [2]. However, high-intensity or prolonged exercise might increase the presence of inflammation-associated gut microbiota (e.g., *Haemophilus*, *Rothia*, *Mucispirillum*, and *Ruminococcus gnavus*) [12], potentially negatively affecting the gut health of athletes who perform frequent high-intensity exercise. Athletes experience high levels of mental and physical stress during the athletic season, while the off-season is characterized by a decrease in training frequency. However, few studies have investigated the impact of these seasonal differences on the gut microbiota of athletes.

This study aimed to compare the gut microbiotas between Japanese male handball players, as athletic subjects (AS), and age-matched, healthy males, as non-athletic subjects (NS). Handball is a sport that demands both speed and endurance, and Japanese male handball is a seasonal sport with distinct athletic and off-season periods. Therefore, gut microbiotas were also compared between the athletic season and the off-season.

## 2. Materials and Methods

### 2.1. Dataset Construction

This study used two datasets, referred to below as datasets 1 and 2, derived from the analysis of fecal samples given to the MORINAGA cohort. The cohort comprised healthy adults with no prior history of cancer, cardiovascular disease, liver disease, or gastrointestinal disease [13]. The participants included players and team staff members of an elite Japanese men’s handball team. In this handball team, the players were neither living in dormitories nor having their meals managed by the team. Informed consent was obtained from all participants. In addition, there were no instructions in this study to intervene in the lifestyle of the participants. This cohort study was approved by the Ethics Committees of the National Institutes of Biomedical Innovation, Health and Nutrition (Osaka, Japan) and the Ethics Committee of the Japan Clinical Research Association (Tokyo, Japan), and all guidelines were followed.

Datasets 1 and 2 were constructed from selected samples of male participants in their 20s and 30s with the aim of matching the age and sex of the players. Dataset 1 comprised samples submitted from November 2021 to March 2022. Dataset 2 comprised samples from three distinct terms: Term 1 (T1), spanning from November 2021 to March 2022; Term 2 (T2), from April 2022 to June 2022; and Term 3 (T3), from July 2022 to November 2022 (Figure 1A). In cases where multiple samples were available within a term, one was randomly selected. The selected samples were divided into two groups, AS and NS, for comparative analysis. Subjects who met the aforementioned criteria were included in this study, resulting in the recruitment of 27 and 14 subjects for datasets 1 and 2, respectively (Figure 1B).

### 2.2. Data Collection

The gut microbiota data were collected in accordance with established protocols, which included fecal sampling, DNA extraction, and 16S rRNA sequencing. Detailed descriptions of these procedures can be found in a previous study [13]. Briefly, fecal samples were collected in containers with guanidine thiocyanate as a preservative solution, supplied by Techno Suruga Laboratory (Shizuoka, Japan). The fecal sample mixtures were then mechanically disrupted using the bead beating method with glass beads. DNA was then extracted using an automated extraction machine (Kurabo Industries, Osaka, Japan). The amplified V3–V4 region of the bacterial 16S rRNA gene was amplified by PCR and sequenced using the paired-end method on an Illumina MiSeq instrument with the MiSeq v3 Reagent Kit (Illumina, Inc., Foster City, CA, USA).

### 2.3. Bioinformatics Analysis

The obtained paired-end FASTQ data from the registered data in the cohort study were trimmed and merged before selection of the amplicon sequence variants (ASVs). The classification and diversity analysis of the ASVs were performed using the QIIME2 software package, version 2022.8 (https://qiime2.org/ accessed on 29 December 2022), as described previously [14]. Taxonomic classification was performed using the naive Bayes classifier trained on Greengenes2, version 2022.10 (https://greengenes2.ucsd.edu/ accessed on 2 February 2023), with a 99% threshold for full-length sequence operational taxonomic units. After the assignment of each ASV to a bacterial species, alpha-diversities were calculated using QIIME2 software. Moreover, principal coordinate analysis (PCoA) based on Bray–Curtis dissimilarity was performed using R software (ver. 4.3.1) using the vegan (version 2.6-4) and ape (version 5.7-1) packages.

### 2.4. Statistical Analysis

All statistical analyses were performed using R software and EZR (ver. 4.2.2) [15]. Data collected as background information were statistically analyzed using Welch’s *t*-test or Fisher’s exact test. The Mann–Whitney U test was used to compare alpha-diversity and the genera of bacteria between AS and NS. When analyzing relative abundance at the genus level, the false discovery rate (FDR) was calculated using the Benjamini–Hochberg procedure after the Mann–Whitney U test, with a significance threshold set at q < 0.05. Permutational analysis of variance (PERMANOVA) for PCoA was conducted in R using the adonis2 function from the vegan package, version 2.6-4. The Friedman test was used for the longitudinal analysis using dataset 2.

### 2.5. Data Availability

The data pertaining to background information, alpha-diversity, and relative abundance at the genus level are presented in Appendix A. DNA sequences corresponding to the 16S rRNA gene data have been deposited in the DDBJ database under accession number DRA017698 (Appendix A).

## 3. Results

### 3.1. Description of Subjects and Background Information in Datasets

The gut microbiota data from 17 AS and 10 NS were included in dataset 1. Dataset 2 comprised data from 5 AS and 9 NS, resulting in a total of 42 unique data entries related to gut microbiota across both datasets (Appendix A). DNA sequencing of the 42 samples yielded 270,660 reads (mean 6444 reads per sample, min 2662, max 12,191), which, after de-noising, resulted in a total of 744 ASVs. In both datasets, AS exhibited significantly higher values for height, weight, and body mass index (BMI) compared with NS. Significant differences were also observed in the duration of activity (“active time”) per day, including sports and/or physical labor (Table 1 and Table 2). Furthermore, the duration of inactivity (“sedentary time”) per day was significantly lower in AS than in NS for dataset 1 (Table 1). A similar result was also observed in dataset 2, but this was not statistically significant (Table 2).

### 3.2. Comparative Analysis of Gut Microbiota Diversity and Composition between AS and NS in Dataset 1

The Shannon (*p* = 0.046) and Chao1 (*p* = 0.025) indices, representing alpha-diversity, were significantly higher in AS compared with NS (Figure 2A). Furthermore, PCoA based on Bray–Curtis dissimilarity metrics also indicated significant differences (*p* = 0.028, Figure 2B).

In terms of the composition at the genus level, the relative abundance of 19 dominant genera (>1%) was compared between AS and NS. The results showed that *Faecalibacterium* (*p* = 0.003, q = 0.024) and *Streptococcus* (*p* = 0.001, q = 0.024) were significantly more abundant in AS than NS (Figure 3). Additionally, *Fusicatenibacter* and *Anaerostipes* tended to be higher in AS than NS, although this difference did not reach statistical significance (*p* < 0.05, but q > 0.05, Appendix A). The detection rate of *Streptococcus* was higher in AS (100%) compared with NS (60%). However, after applying FDR correction, this difference was not statistically significant (*p* = 0.012, q = 0.228, Appendix A).

### 3.3. Longitudinal Analysis of Alpha-Diversity in the Gut Microbiota between the Terms in Dataset 2

To investigate longitudinal changes between the athletic season and off-season, the gut microbiota, as represented by fecal samples collected during the first athletic season (T1), the off-season (T2), and the second athletic season (T3), were compared. A significant longitudinal change in the Shannon index was observed in AS (*p* = 0.022), whereas no such change was observed in NS (*p* = 0.236). The Chao1 index showed similar trends to the Shannon index, but these were not statistically significant (AS: *p* = 0.091 and NS: *p* = 0.368). Furthermore, both the Shannon and Chao1 indices in AS were significantly higher in T1 and T3 (athletic seasons) compared with NS, while no significant difference in both indices was observed between AS and NS in T2 (off-season) (Figure 4A).

### 3.4. Longitudinal Analysis of the Gut Microbiota at the Genus Level between the Terms in Dataset 2

To further examine longitudinal changes in the gut microbiota, the data were analyzed using PCoA based on Bray–Curtis dissimilarity metrics, a significant index between AS and NS in dataset 1. No significant longitudinal changes were observed in either AS or NS (Figure 4B). An attempt was then made to identify genera that showed significant changes between the athletic seasons and the off-season. Among the 202 genera detected in dataset 2, 24 genera (11.9%) in AS showed an inverted V-shaped change, while 58 genera (28.7%) showed a V-shaped change, with higher and lower abundance in the off-season (T2) compared with the athletic seasons (T1 and T3). Although none of these changes were statistically significant, *Collinsella*, *Roseburia*, unclassified *Peptostreptococcaceae* 256921, unclassified *Lachnospiraceae*, and *Dorea A* showed a tendency to change (*p* < 0.05 and q > 0.05, Appendix A). No longitudinal changes were observed in *Faecalibacterium* and *Streptococcus,* which exhibited significant differences between AS and NS in dataset 1 (Appendix A).

## 4. Discussion

The primary objective of this study was to investigate the gut microbiota in Japanese male handball players and to discern any potential changes in their gut microbiota between the athletic season and the off-season. In dataset 1, a comparative analysis of the gut microbiota between AS and NS revealed a significantly distinct gut microbiota in AS compared with NS, as evidenced by the results of PCoA. Specifically, the alpha-diversity in AS was significantly higher than in NS, a finding that aligns with previous studies despite differences in ethnicity and the type of sport [4]. Moreover, AS demonstrated a significantly higher relative abundance of *Faecalibacterium* and *Streptococcus*, while *Fusicatenibacter* and *Anaerostipes* tended to be more abundant in AS than NS. *Faecalibacterium*, *Fusicatenibacter*, and *Anaerostipes,* well-known SCFA-producing genera, have been reported to be highly prevalent in athletes [5,6]. These findings suggest that the gut microbiota of athletes, including handball players, may be enriched in functional pathways such as SCFA production. Indeed, a study employing metagenomics and metabolomics reported that professional rugby players exhibited an abundance of beneficial pathways (e.g., carbohydrate metabolism) and fecal metabolites (e.g., SCFA) compared to healthy controls [16].

In a previous study of gut microbiota changes with training periodization in Japanese elite athletes, *Fusicatenibacter* and *Anaerostipes* tended to increase after training periodization, and a tendency for correlation was also observed between changes in *Fusicatenibacter* and anaerobic power output [17]. *Streptococcus*, which has been reported to increase with high-intensity exercise lasting 95 min or more [2], may be induced by intensive exercise. For example, *S. pyogenes* and *S. pneumoniae* have been suggested to be the major cause of upper respiratory infections, which is a common disease in athletes who undergo heavy training [18,19]. Therefore, further research is necessary to understand both the positive and negative impacts of the gut microbiota in athletes on their physical and mental performance.

The data from dataset 2, despite its limited sample size, were used to investigate the difference in gut microbiota between the athletic season and the off-season. A significant difference was observed in alpha-diversity during the athletic season (T1 and T3) between AS and NS, while no significant difference was observed during the off-season (T2). Although not statistically significant, some genera (e.g., *Roseburia*) showed a tendency to change longitudinally between the athletic season and off-season, similar to alpha-diversity (Appendix A). This suggests potential changes in the gut microbiota of athletes throughout a season. However, due to the limited sample size, these findings should be interpreted with caution and may not be generalizable to a larger population. Further studies with larger sample sizes are needed to confirm these findings and to better understand the changes in gut microbiota throughout an athletic season. Among the limited studies in this area, one report examined the changes in the gut microbiota of professional soccer players during the season [20] and found that the most variation occurred at the start of the season. Taking this into consideration, the beginning of an athletic season could be interpreted as the time when the modified gut microbiota associated with the off-season is returning to the state associated with the athletic season. Considering the impact of gut microbiota on performance, alterations in the gut microbiota during the off-season could potentially impact athletes’ health, recovery, and performance as they transition back to the athletic season. For instance, if the shift in the gut microbiota from the off-season to athletic season is delayed, it could potentially affect performance at the start of the athletic season.

Interviews were conducted with the players, staff, and managers of the handball team, including the participants in this cohort study, to explore the possible reasons for changes between the athletic season and off-season. The interviews primary focused on diet and physical activity, which are well-established factors affecting the gut microbiota [1,2]. Most of the interviewed athletes reported that their dietary habits and dietary supplement (e.g., protein supplement) intake remained consistent throughout the year, both during the athletic season and the off-season. However, we acknowledge that these reports are subjective, and we were unable to obtain objective data to confirm the consistency of their diet throughout the year. Therefore, we could not definitively quantify the impact of diet on the observed changes in the gut microbiota between the off-season and the athletic season.

A systematic review investigating the impact of exercise on the gut microbiota reported that an exercise frequency of two to three times per week resulted in no significant changes in alpha-diversity, while an exercise frequency of four to five times per week and more led to an increase in alpha-diversity [2]. Furthermore, the impact of an 8-week exercise intervention on the gut microbiota in sedentary subjects was reported to increase alpha-diversity, which returned to pre-intervention levels 3 weeks after stopping the exercise intervention [21]. However, there have been few studies that provide comparative evaluations of the effects of different types of exercise (e.g., anaerobic versus aerobic exercise) on the gut microbiota. While higher levels of physical activity and cardiorespiratory fitness have been reported to be positively associated with alpha-diversity in the gut microbiota [22], further research is needed to fully understand these influences. Despite the limited evidence available, it was postulated that the observed changes during the off-season might be primarily attributed to a decrease in physical activity rather than alterations in diet or nutritional status within this team. However, substantiating this hypothesis presents a challenge due to the myriad of complex factors, beyond diet and physical activity, that can influence the gut microbiota [13,22]. To elucidate the lifestyle differences between athletic seasons and off-seasons that influence the gut microbiota, comprehensive additional analyses on each factor are warranted.

This study had some limitations. Specifically, a small sample size was analyzed. This was particularly noticeable because dataset 2 only included samples that were submitted in all three terms. To confirm the new findings in this study, it will be necessary to conduct larger trials in the future focusing on the time before and after the athletic season. To investigate the mechanisms of impact on the gut microbiota throughout a season, data on not only the gut microbiota, but also on dietary intake and physical activity, will need to be collected. Data also need to be collected because lifestyle changes during the off-season vary depending on the team. Additionally, it is notably important to assess how these changes in the gut microbiota throughout a season influence performance and health in athletes. The gut microbiota, particularly the metabolites derived from it (e.g., SCFAs), have been demonstrated to potentially influence the intestinal environment, immune system, and systemic metabolism, which could subsequently impact athletic performance and health [7,23]. Specifically, SCFA can enhance an athlete’s immunity, exert anti-inflammatory effects, and provide additional energy substrates during endurance exercise [24]. Interestingly, a recent study also reported that gut microbiota-derived metabolites can improve exercise performance by stimulating sensory nerves in the gut, thereby enhancing activity in the brain region controlling motivation during exercise [25]. Given these findings, understanding the gut microbiota in athletes is critical to maintaining their performance and conditioning. Therefore, by accumulating fundamental evidence such as this study, we hope to contribute to optimizing the nutritional strategy for athletes throughout the year, including the off-season.

## 5. Conclusions

This study, being the first of its kind to investigate the gut microbiota in handball players, has revealed significant differences in the gut microbiota between AS and NS. Notably, AS exhibited higher alpha-diversity during the athletic season, which decreased during the off-season. This suggests that the gut microbiota is likely to be influenced during the off-season when lifestyle factors, such as physical activity, are potentially changing. If the hypothesis was considered that the gut microbiota can influence an athlete’s performance, our findings underscore the importance of managing the gut microbiota during the off-season. In particular, it may be crucial to guide the gut microbiota back to its balanced state during the transition from the off-season to the athletic season. This could potentially ensure a successful start to the season, optimizing athletes’ performance. Further research is needed to validate these findings and to explore the potential strategies for managing the gut microbiota in athletes, which could open new avenues for enhancing athletic performance and health.

## Figures and Tables

**Figure 1 microorganisms-12-00781-f001:**
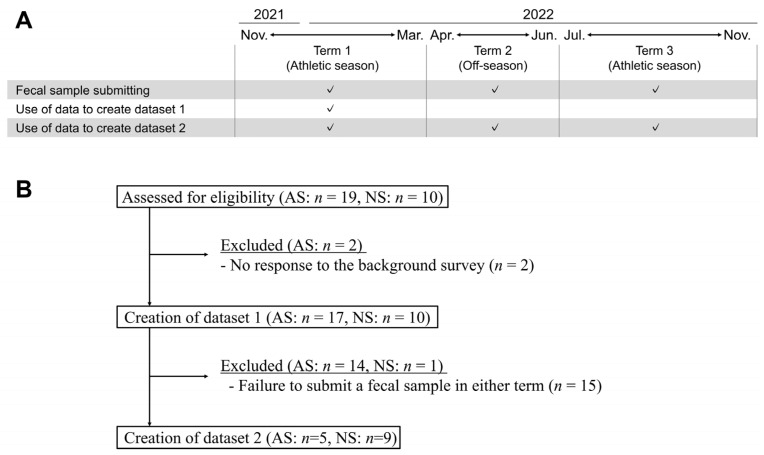
Study scheme and the generation of datasets. (**A**) Fecal samples were submitted from three distinct terms: Term 1 (T1), spanning from November 2021 to March 2022; Term 2 (T2), from April 2022 to June 2022; and Term 3 (T3), from July 2022 to November 2022. Fecal samples could be submitted more than once throughout this trial. In cases where multiple samples were available within a term, one was randomly selected. Two datasets were created: dataset 1 comprised samples submitted from T1; dataset 2 comprised samples from the three distinct terms. (**B**) Data from subjects who met the criteria were selected, and 27 and 14 subjects were finally recruited for datasets 1 and 2, respectively. Abbreviations: athletic subjects (AS), non-athletic subjects (NS).

**Figure 2 microorganisms-12-00781-f002:**
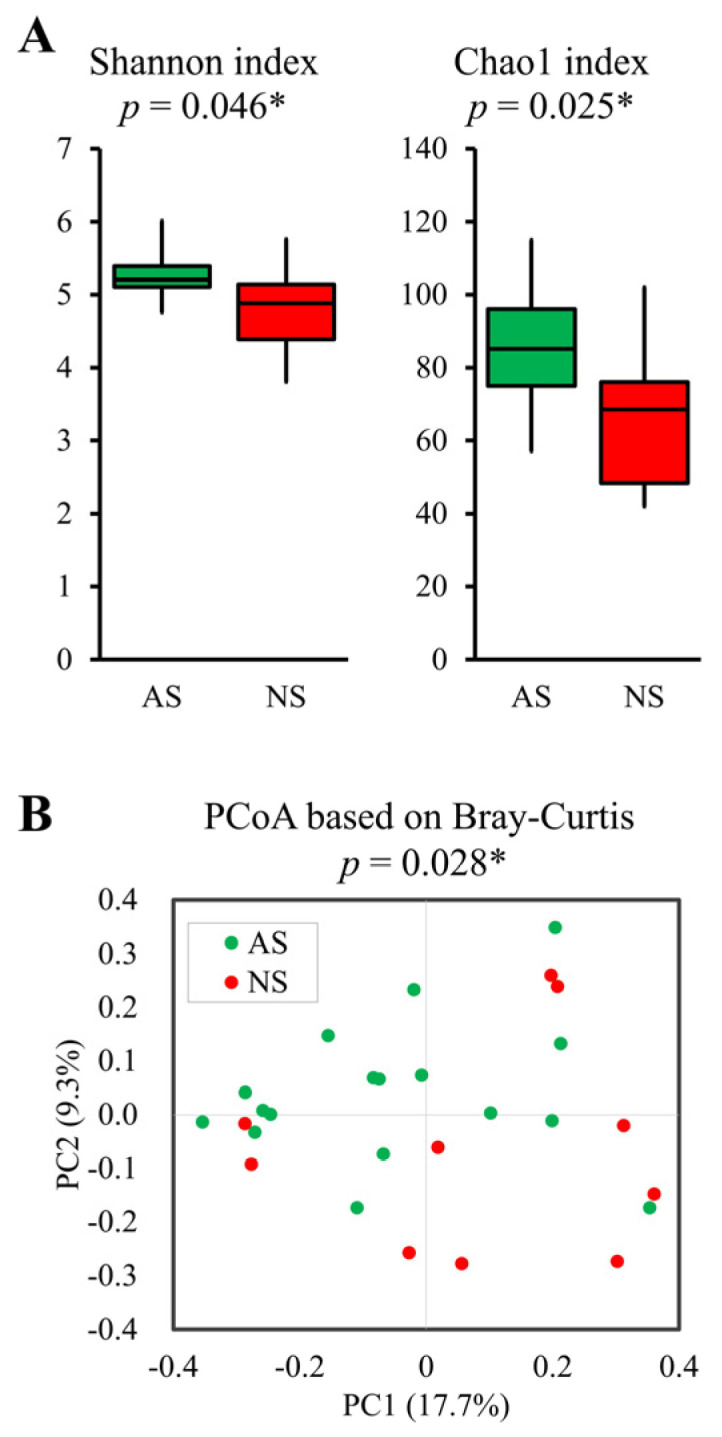
Comparative analysis of alpha- and beta-diversity between AS and NS. Alpha- (**A**) and beta-diversity plots (**B**) to visualize the differences in gut microbiota between AS (*n* = 17, green) and NS (*n* = 10, red). Alpha-diversity was measured using the Shannon and Chao1 indices (**A**). Box plots show the median, as well as the lower and upper quartiles. Whiskers represent the minimum and maximum spread. PCoA plots show the beta-diversity with Bray–Curtis dissimilarity (**B**). Each dot represents an individual sample. Statistical differences in alpha- and beta-diversity were analyzed for significance using the Mann–Whitney U test and PERMANOVA, respectively (* *p* < 0.05).

**Figure 3 microorganisms-12-00781-f003:**
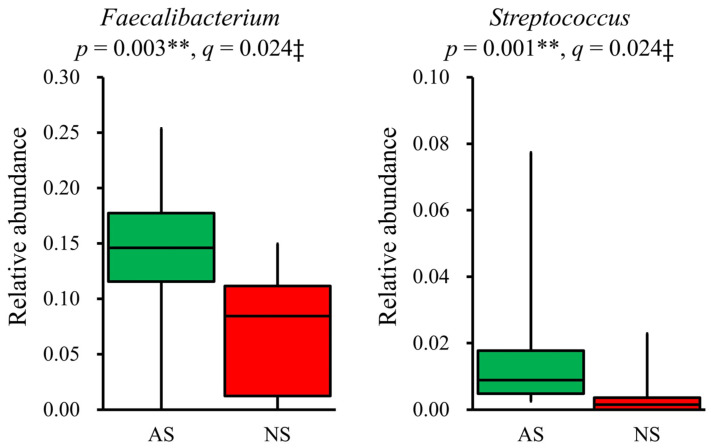
Significant differences in genera present in the gut microbiota between AS and NS. Among 19 dominant genera (>1%), *Faecalibacterium* and *Streptococcus* showed significant differences between AS (*n* = 17, green) and NS (*n* = 10, red), as presented by box plots. Statistical analysis was performed using the Mann–Whitney U test and the Benjamini–Hochberg procedure, with *p* < 0.05 and q < 0.05 considered statistically significant: ** *p* < 0.01, ‡ q < 0.05.

**Figure 4 microorganisms-12-00781-f004:**
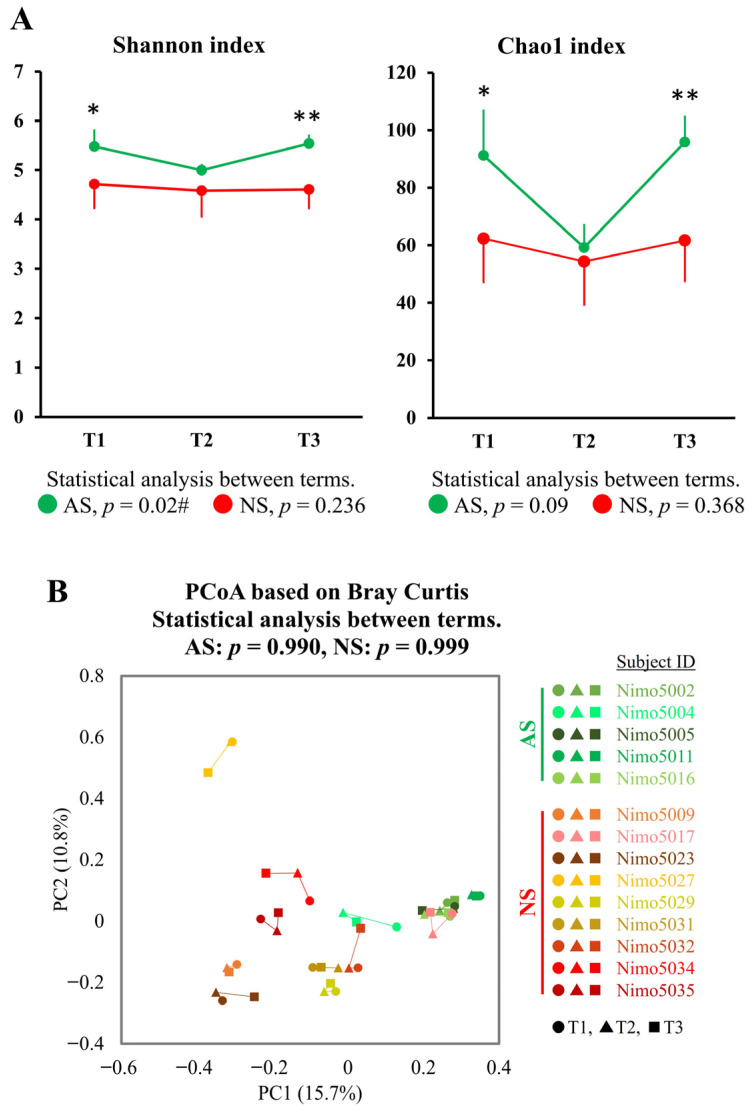
Longitudinal analysis of alpha- and beta-diversity between the athletic season and the off-season. (**A**) Alpha-diversity in AS (*n* = 5, green) and NS (*n* = 9, red) was measured using the Shannon and Chao1 indices. The data are expressed as the mean and the standard deviation. Statistical significance was assessed using the Mann–Whitney U test and the Friedman test for comparisons between groups (* *p* < 0.05, ** *p* < 0.01) and between terms (# *p* < 0.05), respectively. (**B**) PCoA plots showing the beta-diversity with Bray–Curtis dissimilarity. The samples are represented by individual dots, with shades of green representing AS (*n* = 5: Nimo5002, Nimo5004, Nimo5005, Nimo5011, and Nimo5016), shades of red representing NS (*n* = 9: Nimo5009, Nimo5017, Nimo5023, Nimo5027, Nimo5029, Nimo5031, Nimo5032, Nimo5034, and Nimo5035), circles (●) representing T1, triangles (▲) representing T2, and squares (■) representing T3. The differences between the terms were analyzed using PERMANOVA. Abbreviations: T1, Term 1 (athletic season); T2, Term 2 (off-season); T3, Term 3 (athletic season).

**Table 1 microorganisms-12-00781-t001:** Background information for dataset 1.

		AS	NS	*p*-Values	
		(*n* = 17)	(*n* = 10)		
Height	cm	182.6 ± 6.8	176.6 ± 7.9	*p* = 0.073	Welch’s *t*-test
Body weight	kg	87.5 ± 8.4	70.6 ± 10.0	*p* < 0.01 **	Welch’s *t*-test
BMI	kg/m^2^	26.2 ± 1.4	22.6 ± 2.3	*p* < 0.01 **	Welch’s *t*-test
Age	Twenties	13	4	*p* = 0.101	Fisher’s exact test
	Thirties	4	6		
Active time per day (e.g., sports and/or physical labor)	less than 30 min	0	6	*p* < 0.01 **	Fisher’s exact test
	30 min–3 h	12	4		
	3 h or more	5	0		
Sedentary time per day	less than 3 h	2	1	*p* = 0.014 *	Fisher’s exact test
	3–8 h	15	5		
	8 h or more	0	4		

Statistical analysis was performed using Welch’s *t*-test and Fisher’s exact test: * *p* < 0.05, ** *p* < 0.01. Abbreviations: athletic subjects (AS); non-athletic subjects (NS).

**Table 2 microorganisms-12-00781-t002:** Background information for dataset 2.

		AS	NS	*p*-Values	
		(*n* = 5)	(*n* = 9)		
Height	cm	185.2 ± 5.5	175.6 ± 8.1	*p* = 0.022 *	Welch’s *t*-test
Body weight	kg	90.2 ± 5.8	68.7 ± 9.1	*p* < 0.01 **	Welch’s *t*-test
BMI	kg/m^2^	26.3 ± 1.4	22.2 ± 2.3	*p* < 0.01 **	Welch’s *t*-test
Age	Twenties	3	3	*p* = 0.59	Fisher’s exact test
	Thirties	2	5		
Active time per day (e.g., sports and/or physical labor)	less than 30 min	0	6	*p* < 0.01 **	Fisher’s exact test
	30 min–3 h	2	3		
	3 h or more	3	0		
Sedentary time per day	less than 3 h	0	1	*p* = 0.15	Fisher’s exact test
	3–8 h	5	4		
	8 h or more	0	4		

Statistical analysis was performed using Welch’s *t*-test and Fisher’s exact test: * *p* < 0.05, ** *p* < 0.01. Abbreviations: athletic subjects (AS); non-athletic subjects (NS).

## Data Availability

Data are contained within Appendix A.

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
