# Peer review of "An Exploratory Study on Seasonal Variation in the Gut Microbiota of Athletes: Insights from Japanese Handball Players"

_microorganisms, 2024, doi:10.3390/microorganisms12040781_

Round 1

Reviewer 1 Report (Previous Reviewer 2)

Comments and Suggestions for Authors

The study investigates the seasonal variation of gut microbiota between athletic subjects (AS) and non-athletic subjects (NS) and examines changes between in-season and off-season periods. Utilizing 16S rRNA sequencing for gut microbiota profiling, the study reveals a distinct gut microbiota composition in athletes compared to non-athletes, highlighting significant differences in alpha diversity and the relative abundance of genera such as Faecalibacterium and Streptococcus. The article is interesting, but is limited by its small, sport-specific sample size and lack of detailed dietary analysis, which together restrict the generalizability of its findings and hinder a comprehensive understanding of diet's impact on observed microbiota variations.

Main weaknesses:

-      The study's limited sample size, especially in dataset 2, could impact the generalizability of the findings.

-      The paper mentions diet as a significant factor affecting gut microbiota, it does not provide detailed dietary data or its variation across seasons.

How might dietary variations throughout the athletic and off-seasons influence the observed changes in gut microbiota? This question seeks to address the notable gap regarding diet's role in modulating gut microbiota composition in athletes.

-      While the study attempts to control for confounders, the inherent variability in lifestyle, diet, and training intensity across athletes and seasons might still influence the gut microbiota beyond the measured variables.

-      There are identified associations between gut microbiota composition and athletic seasonality but offers limited insights into the underlying mechanisms driving these changes.

What specific physical activity patterns or training regimens correlate most strongly with beneficial changes in gut microbiota among athletes?

How do changes in gut microbiota composition during the off-season impact athletes' health, recovery, and performance as they transition back to the athletic season?

Author Response

We would like to express our sincere gratitude for your insightful comments and constructive feedback on our study. Your input has greatly contributed to the improvement of our paper. Followings provides the specific changes against your comments, one by one. For ease of reference, all revisions in the manuscript have been highlighted in red. We hope that the revisions and our responses to your comments have sufficiently addressed your concerns.

  • The study's limited sample size, especially in dataset 2, could impact the generalizability of the findings.

>>> We fully acknowledge and appreciate your concerns regarding the sample size. We understand that this is a critical factor in ensuring the validity and reliability of any study. As such, we have addressed this as a key limitation in our discussion section. Furthermore, to emphasize the exploratory nature of our study due to the small sample size, we had titled our paper as follows: "An Exploratory Study on Seasonal Variation in the Gut Microbiota of Athletes: Insights from Japanese Handball Players". In response to your feedback, we have expanded our discussion on the potential implications of our small sample size (p8, Lines 287-290). Despite these limitations, we believe our study provides valuable preliminary insights into the seasonal variation in athletes' gut microbiota, a topic that has not been extensively investigated in previous research. If we may add, we have ensured that our data interpretation and subsequent discussion are based on results derived from appropriate statistical analysis, in order to avoid over-speculation.

  • The paper mentions diet as a significant factor affecting gut microbiota, it does not provide detailed dietary data or its variation across seasons. How might dietary variations throughout the athletic and off-seasons influence the observed changes in gut microbiota? This question seeks to address the notable gap regarding diet's role in modulating gut microbiota composition in athletes.

>>>We appreciate your insightful comment regarding the potential influence of diet on gut microbiota. As indicated on p9, Lines 305-307, we conducted interviews with the team and players, which suggested that their diet remained consistent throughout the year, encompassing both the competitive season and the off-season. However, we acknowledge that we were unable to obtain objective data to demonstrate the extent of dietary variation, if any, and thus, we could not definitively quantify the impact of diet on the observed changes in gut microbiota between the off-season and the athletic season. We have added this point to our discussion section to address this limitation and highlight the need for future research to consider dietary factors when investigating changes in athletes' gut microbiota across different seasons (p9, Lines 307-311).

  • While the study attempts to control for confounders, the inherent variability in lifestyle, diet, and training intensity across athletes and seasons might still influence the gut microbiota beyond the measured variables.

>>> We appreciate your insightful comment regarding the inherent variability in lifestyle, diet, and training intensity across athletes and seasons. We acknowledge that these factors could introduce additional complexity into our findings. While we attempted to control for known confounders, we recognize the challenge of accounting for all potential influences. This limitation underscores the complexity of studying gut microbiota. Your feedback is valuable and highlights important considerations for future research.

There are identified associations between gut microbiota composition and athletic seasonality but offers limited insights into the underlying mechanisms driving these changes. What specific physical activity patterns or training regimens correlate most strongly with beneficial changes in gut microbiota among athletes?

>>> Thank you for your insightful comments. In our understanding, it was remain under considerations what specific physical activity patterns or training regimens correlate "most strongly" with beneficial changes in gut microbiota among athletes. However, higher levels of physical activity and cardiorespiratory fitness have been reported to be positively associated with alpha-diversity in the gut microbiota. Therefore, we added above explanations to expand the discussion in p9, Lines 318-323.

  • How do changes in gut microbiota composition during the off-season impact athletes' health, recovery, and performance as they transition back to the athletic season?

>>>Thank you for your insightful question. As you suggested, off-season gut microbiota changes should significantly impact athletes' health and performance. While our study did not directly investigate these impacts, we recognize their potential significance. Therefore, we have expanded our discussion to include a consideration of how changes in gut microbiota during the off-season might influence athletes' health and performance as they transition back to the athletic season (p8, Lines 295-300).

Once again, we are grateful for your valuable comments and queries that have allowed us to strengthen our manuscript. We have made every effort to incorporate your feedback and hope that these revisions meet your approval for our submission.

Reviewer 2 Report (New Reviewer)

Comments and Suggestions for Authors

The microbiota can affect the body's physiological parameters, potentially leading to health benefits or contributing to the development of diseases. The relationship between the microbiota and human physical activity has been established. Different sporting disciplines have been found to have varying human gut microbiota, suggesting that certain sports may promote specific microbial ecosystems. The microbiome of 'elite athletes' exhibits peculiarities in terms of species composition and strain properties. It is well-established that regular intake of probiotics, consisting of specific strains of probiotic bacteria, over a period of several weeks to months, is associated with improved sports performance and faster recovery of athletes' physical condition.  The manuscript offers valuable insights for experts in microbiology and sports medicine.

The manuscript is well-structured, including all the necessary sections, such as Introduction, Materials and Methods, Results and Discussion, and Conclusion.  The reference list comprises 25 sources, with 64% of them published between 2020 and 2024. Each section is clearly written and easy to understand. The manuscript is written in a clear and concise manner. Despite the positive reviews, the authors have some questions, as discussed below.

1.      How do you think that eating the foods studied affected the results obtained?

2.      Could possible climatic (seasonal) changes have an impact on the biochemical parameters of the organism and, through them, on the microbiota?

Author Response

We appreciate for the thoughtful and constructive feedback you provided regarding our manuscript. Your comments have been instrumental in deepening our understanding of the findings. We have addressed your comments as follows and hope that our responses will clarify any concerns you may have.

-How do you think that eating the foods studied affected the results obtained?

>>> We agree with your point that diet could potentially have a significant impact on our results. As indicated on p9, Lines 305-307, we conducted interviews with the team and players, which suggested that their diet remained consistent throughout the year, encompassing both the competitive season and the off-season. However, we were unable to obtain objective data to demonstrate the extent of dietary variation, if any, and thus, we could not definitively quantify the impact of diet on the observed changes in gut microbiota between the off-season and the competitive season.

-Could possible climatic (seasonal) changes have an impact on the biochemical parameters of the organism and, through them, on the microbiota?

>>>As you mentioned, we also think that climatic (seasonal) changes may have an impact on the gut microbiota. In fact, a longitudinal study investigating the gut microbiota of the Hutterites was reported to observe seasonal variations in gut microbiota composition between summer and winter [Davenport ER, et al. PLoS One. 2015]. However, in this study, we compared the gut microbiota between athletes and non-athletes and identified differences between the two groups. We observed that the gut microbiota in non-athletes remained stable, while that in athletes fluctuated throughout the year. Therefore, we believe that the shift in gut microbiota in athletes is not due to climate or seasonal changes, but rather due to the transition between the athletic season and the off-season.

Thank you once again for your valuable feedback and for considering our paper.

This manuscript is a resubmission of an earlier submission. The following is a list of the peer review reports and author responses from that submission.

Round 1

Reviewer 1 Report

Comments and Suggestions for Authors

The study compares gut microbiotas between athletic subjects (AS) and non-athletic subjects (NS) and investigates variations between athletic and off-season periods. While this topic has been explored in previous research, the novelty of this study seems somewhat limited in terms of both writing style and research perspective.

The identification of microbial abundances within the two groups reveals significant heterogeneity, particularly within the AS group. This suggests that environmental factors influencing gut microbiota variation may not have been adequately controlled within the AS group. It would be beneficial for the manuscript to detail any dietary or lifestyle restrictions imposed on the study groups, as this could greatly impact the interpretation of the results. Additionally, the discussion of relevant research findings should be incorporated into the manuscript.

Furthermore, the current study has a relatively small sample size and lacks in-depth exploration and analysis of the existing data. Therefore, based on the current state of the manuscript, I do not believe it is suitable for publication.

Comments on the Quality of English Language

no

Reviewer 2 Report

Comments and Suggestions for Authors

The study addresses an emerging and highly relevant area of research, exploring the gut microbiota's role in athletic performance and health. The focus on seasonal variations and the comparison between athletic and non-athletic subjects add valuable insights to the field. The use of longitudinal and comparative analysis provides a robust framework for understanding the dynamic nature of the gut microbiota in response to athletic training and seasonal changes. However, the relatively small sample size and the specificity of the study population (Japanese handball players) may limit the generalizability of the findings. Expanding the study to include a more diverse group of athletes from various sports and regions could enhance the applicability of the results.

My comments are listed below:

I recommend the authors to avoid using the first person plural

The study reports significant differences in the abundance of specific genera between AS and NS. Considering the small sample size, how robust are these findings against the potential for overfitting or statistical anomalies?

Given the dynamic nature of the gut microbiota and the possible effects of diet and exercise, no significant longitudinal changes were found at the genus level in either AS or NS. How do the authors interpret this finding, and what might it imply about the stability or resilience of the gut microbiota composition in response to seasonal athletic activities?

The authors discuss the significant differences in alpha-diversity and some genera tendencies between the athletic and off-season, suggesting that these changes might be attributed more to variations in physical activity rather than diet. Considering the complexity of factors influencing gut microbiota composition, including stress, sleep, and environmental factors, how do the authors differentiate the impact of physical activity from these other variables?

While the results are well-documented, a more detailed discussion on the potential mechanisms by which variations in gut microbiota affect athletic performance could be beneficial. This could include a deeper dive into the metabolic pathways involved and how they might interact with exercise physiology and nutrition.